# A Simple and Efficient Baseline for Data Attribution on Images

## Abstract

Data attribution methods play a crucial role in understanding machine learning models, providing insight into which training data points are most responsible for model outputs during deployment. However, current state-of-the-art approaches require a large ensemble of as many as 300,000 models to accurately attribute model predictions. These approaches therefore come at a high computational cost, are memory intensive, and are hard to scale to large models or datasets. In this work, we focus on a minimalist baseline that relies on the image features from a pretrained self-supervised backbone to retrieve images from the dataset. Our method is model-agnostic and scales easily to large datasets. We show results on CIFAR-10 and ImageNet, achieving strong performance that rivals or outperforms state-of-the-art approaches at a fraction of the compute or memory cost. Contrary to prior work, our results reinforce the intuition that a model's prediction on one image is most impacted by visually similar training samples. Our approach serves as a simple and efficient baseline for data attribution on images.

## 1 Introduction

The effectiveness of a machine learning system's performance hinges on the quality, diversity, and relevance of the data it is trained on (Halevy et al., 2009; Sun et al., 2017). In various real-world machine learning systems, for example in healthcare or finance, we often ask questions like, "Which training samples influenced this prediction?" or "How sensitive is this model's prediction to changes in the training data?" Counterfactual insights enable us to assess the impact of hypothetical changes in the data distribution, which in turn helps us understand the basis of the model's decisions and how to change the decision in the event of an error.

These questions motivate research on *data attribution* methods, which focus on understanding which data points most strongly influence a model's outputs. Data attribution methods have been applied to applications such as debugging model biases (Ilyas et al., 2022; Park et al., 2023; Shah et al., 2023), fairness assessment (Black & Fredrikson, 2021), and active learning (Liu et al., 2021).

In principle, data attribution can be done perfectly by a brute-force leave-$k$-out strategy; simply train the model from scratch many times, removing $k$ data points each time. The user can then examine the impact of each data point by examining how the corresponding ablated model differs from the original. Clearly, this procedure is intractable for any realistic problem as there are innumerable subsets, and training even a single machine learning model can be almost prohibitively expensive. The goal of data attribution research therefore is to approximate this gold standard metric as closely as possible while simultaneously using as little computation as possible. As such, the field of data attribution is all about trade-offs between accuracy, runtime, and memory.

Existing data attribution approaches gain insights into model behaviors by scraping information from the learning algorithm, such as logits (Ilyas et al., 2022) or gradients (Koh & Liang, 2017; Park et al., 2023). Despite this, these techniques still require re-training multiple models on different data subsets, or other compute and memory intensive strategies for better efficacy (Ilyas et al., 2022; Feldman & Zhang, 2020; Koh & Liang, 2017; Park et al., 2023). Current data attribution approaches quickly become intractable as datasets become larger (Basu et al., 2021; Park et al., 2023) and applications become more realistic, such as attribution for LLMs (Grosse et al., 2023).

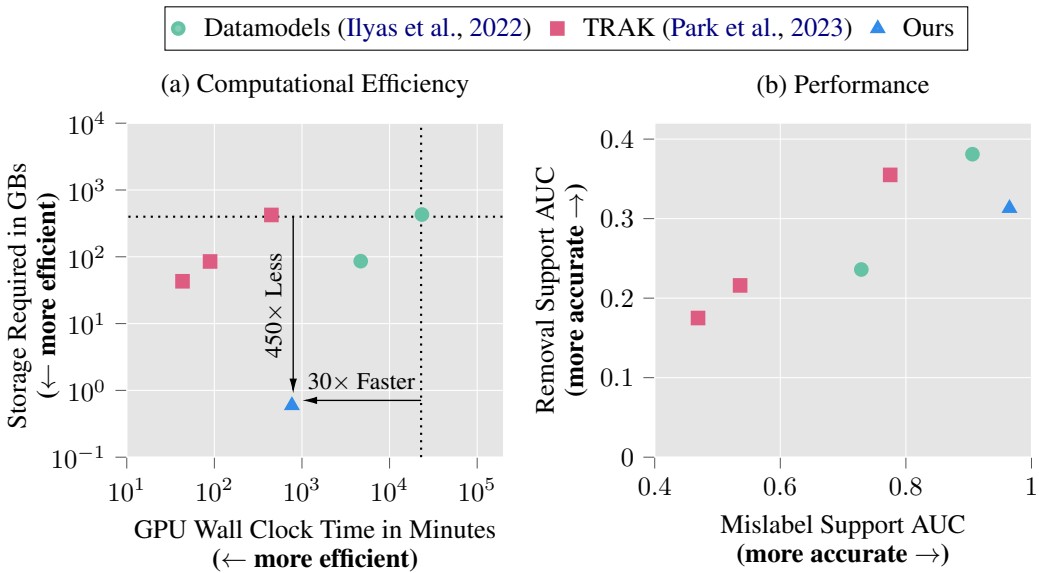

Figure 1: **Our proposed baseline approach for data attribution achieves high performance while improving computational efficiency**. Figure (a) shows the wall-clock time on an RTX A6000 GPU on the x-axis and memory requirements in GBs on the y-axis respectively (see Appendix A.1 for details). Figure (b) shows performance on two metrics measuring the method's accuracy to make counterfactual predictions (details about the metrics are discussed in Section 2.1.)

In this work, we present a simple approach that outperforms the current state of the art in terms of compute-accuracy trade-offs, and often in terms of raw performance numbers as well. Given a test image, we use the feature space of a single self-supervised model to retrieve similar images, revealing a compelling association between data attribution and *visual similarity*. In contrast to existing methods that involve unwieldy model ensembles and extensive computation, our approach shifts the spotlight directly onto the data. Building on prior research, we focus on counterfactual prediction (Ilyas et al., 2022; Park et al., 2023) for evaluating data attribution techniques. Based on the intuition that data inherently shapes model behavior, our method does not use any information about the model training process, and yet still rivals the performance of state-of-the-art approaches that do, while using a tiny fraction of the computational resources. Our work shows that, contrary to previous work (Ilyas et al., 2022; Park et al., 2023), feature representations can serve as a robust baseline for data attribution methods. Our code will be made available with this submission as supplementary material.

## 2 PROBLEM SETTING

We first define our notation and then discuss evaluation criteria used for data attribution approaches. We borrow notation and evaluation criteria from Ilyas et al. (2022) and Park et al. (2023).

**Notation**: Let $S = \{z_1, z_2, \ldots z_n\}$ denote a set of training samples. Each sample $z_i \in S$ represents $z_i = (x_i, y_i)$, where $x_i$ signifies the input image and $y_i$ represents the associated ground truth label. We use $z_t$ to denote an arbitrary evaluation sample not present in the training set.

We denote a data attribution approach as a function $\tau(z, S) \in \mathbb{R}^n$. This function operates on any sample $z$ and a training set $S$, generating a score for each sample within the set $S$. These scores highlight the relative positive or negative impact of individual training samples on the classification of the input sample $z$.

### 2.1 EVALUATING ATTRIBUTION METHODS

Obtaining ground truth for data attribution has been a challenging problem. Several works have focused on evaluating data attribution methods using alternatives such as Shapley values or leave-

one-out influences (Koh & Liang, 2017; Lundberg & Lee, 2017; Jia et al., 2021). These approaches however do not scale beyond modest dataset sizes. An alternate line of work evaluates the utility of attribution methods for auxiliary tasks such as active learning or identifying mislabeled or poisoned data samples (Liu et al., 2021; Jia et al., 2021).

Recent research primarily concentrates on evaluating the performance of data attribution methods through the lens of their capacity to provide accurate counterfactual predictions (Park et al., 2023; Ilyas et al., 2022). While these metrics can be computationally demanding, they represent a straightforward, yet valuable, proxy for assessing the effectiveness of attribution approaches. In our work, we replicate the approach presented in Ilyas et al. (2022) and focus on **data brittleness**. Data brittleness metrics leverage attribution techniques to answer the following question: "*To what extent are model predictions sensitive to modifications in the training data?*" Hence, these metrics serve as a means of estimating counterfactual scenarios. To quantify data brittleness, we focus on two distinct types of data support for a validation sample $z_t$. We explain these below:

**Data Removal Support:** The smallest subset $R_r$, that when removed from the training set $S$, causes an average training run of the model to misclassify $z_t$.

**Data Mislabel Support:** The smallest training subset $R_m$, whose mislabeling causes an average training run of the model to misclassify $z_t$. For each training sample in $R_m$, we change the labels to the second-highest predicted class for $z_t$.

Intuitively, a better data attribution approach should be able to find a smaller subset of training samples that can misclassify $z_t$. We estimate these metrics over a set of validation samples and plot the cumulative distribution (CDF), which represents the probability that a sample's label can be flipped as a function of the data subset size. In Fig. 1, we compare the Area Under Curve (AUC) of the CDF for the metrics described above across our approach and other attribution methods.

For a validation sample $z_t$ and a data attribution approach $\tau(z, S)$, we rank the training samples based on decreasing order of positive influence on $z_t$. Then, based on the ranking, we iteratively select and modify a subset of training data. We perform this search, over different subsets to compute the smallest training subset that can cause $z_t$ to be misclassified. Naively, checking all possible subsets would be computationally expensive. Ilyas et al. (2022) check only subsets with certain discrete sizes to keep costs manageable. We instead propose to perform a **bisection search** to approximate the search for the smallest subset, yielding more accurate results. The bisection search approximation is supported by the observation that several data attribution approaches are additive (Park et al., 2023). The exact algorithm and details are discussed in Appendix A.4.

Linear Datamodeling Score (LDS) is another related metric used for the evaluation of data attribution methods (Ilyas et al., 2022; Park et al., 2023). Note that the LDS metric focuses on counterfactual predictions for *arbitrary* changes in training data. In contrast, data brittleness serves to quantify the accuracy of counterfactual predictions using *targeted* changes to training data based on a specific validation sample. Thus, the latter metric serves as a better proxy for the data attribution method's usefulness as a debugging tool. In this work, we emphasize performance on data brittleness and provide results for the LDS metric in Appendix A.5.

## 3  OUR APPROACH & BASELINES

Our approach utilizes the feature space of a neural network to extract features from a validation sample $z_t$ and each training sample in $S$. We then compute the attribution scores by measuring the distance in feature space between $z_t$ and each training sample in $S$. Prior works have tried similar approaches and claimed them to be ineffective for counterfactual estimation (Park et al., 2023; Ilyas et al., 2022). In the next sections, we describe the details of our approach and discuss our baselines.

### 3.1  DESIGN CHOICES

Our data attribution approach relies on the comparison of image embeddings, and in doing so, we make decisions regarding the choice of feature extractor, the subset of training images to compare, and the distance function.

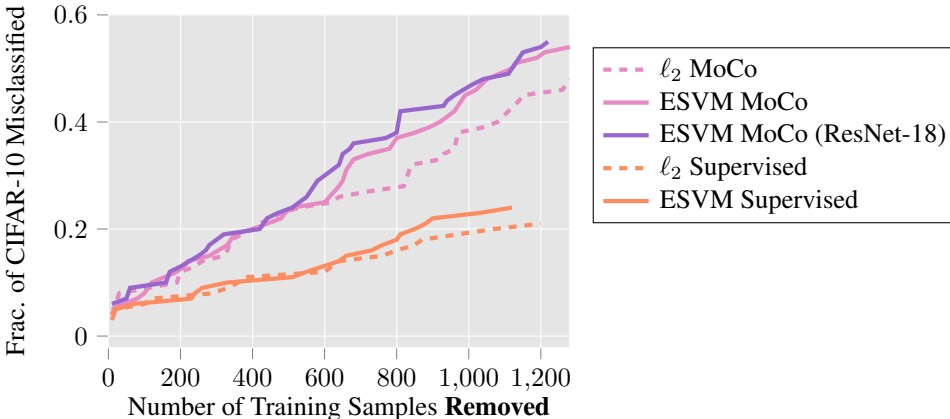

Figure 2: **Self-supervised features are more effective than supervised and are best compared using an ESVM**. Self-supervised features from MoCo can be used to find smaller data support than standard supervised features. For a larger fraction of test samples, ESVM distance is more effective than $\ell_2$ distance at ranking train images to select smaller data removal support.

**Feature extractor.** We find that the learning paradigm used to train a feature extractor heavily influences the estimation of data support. For example, embeddings from a ResNet-9 trained using a self-supervised learning objective (MoCo, (He et al., 2020)) can be used to find smaller support sets than the same model trained in a supervised manner (See $\ell_2$ MoCo vs $\ell_2$ Supervised in Fig. 2). With the exception of DINO (Caron et al., 2021), all self-supervised feature extractors perform better than their supervised counterpart (see Appendix A.2 Fig. 8). We found that MoCo features outperform other self-supervised approaches in both data removal support and mislabeling support scenarios, leading us to select a MoCo model as our preferred feature extractor.

**Subset of train images.** In Appendix A.3 Fig. 10, we show that choosing a support set from training images of class the same class $y$ as the target $z_t = (x, y)$ is critical, *i.e.* given a target image of an airplane, we only rank airplane training images.

**Distance function.** When measuring the distance between two embeddings, Euclidean distance ($\ell_2$) is a common choice (Ilyas et al., 2022; Park et al., 2023). Cosine distance and Mahalanobis distance have also been used to measure similarity, but these were found to perform similarly to Euclidean distances in previous work (Hanawa et al., 2021; Ilyas et al., 2022; Park et al., 2023).

However, we find that measuring distance as distance to the hyperplane of an Exemplar SVM (ESVM) improves image similarity (Malisiewicz et al., 2011). To compute this metric, we train a linear SVM using one positive sample (the target embedding) and treat all other samples (the remaining embeddings of the same class) as negative samples. In this way, the decision boundary, and consequently the distance function, is defined largely by unique dimensions of the target with respect to all embeddings of the same class. In Fig. 2, we demonstrate how using distance to the hyperplane of an ESVM yields better removal support estimates than $\ell_2$ distance.

## 3.2 BASELINES

**Datamodels** (Ilyas et al., 2022): In the *Datamodeling* framework, the end-to-end training and evaluation of deep neural networks is approximated with a parametric function. Surprisingly, optimizing a linear function is enough to predict model outputs reasonably well, when given a training data subset. By collecting a large dataset of subset-output pairs, Ilyas et al. (2022) demonstrate that such a linear mapping can accurately predict the correct-class margin. Among other use-cases, these Datamodels are shown to be effective at counterfactual predictions and identifying visually similar train-test samples. But Datamodeling is prohibitively expensive, requiring the training of hundreds of thousands of models (300,000 in the original work) to generate optimal subset-output data. Unfortunately, this limitation makes Datamodeling intractable for all but small toy problems.

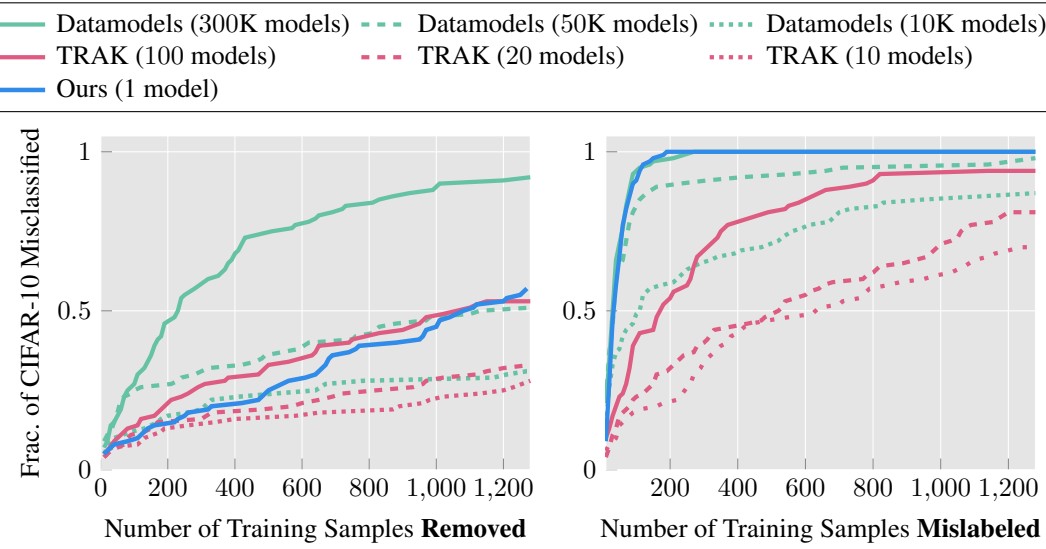

Figure 3: **Our baseline approach uses only a single model and outperforms TRAK and Datamodels using 20 and 10,000 models for data brittleness metrics.** We estimate data removal and data mislabel support for 100 random CIFAR-10 test samples using a Resnet-9 model and plot the cumulative distribution using our approach and other baselines. The number of models used by each approach is also shown. For data removal support, using only a single model our proposed approach outperforms TRAK (Park et al., 2023) using 20 models and Datamodels (Ilyas et al., 2022) using 10,000 models. For data mislabel support, we outperform TRAK using 100 models and perform equivalent to Datamodels using 300,000 models.

**TRAK** (Park et al., 2023): By approximating models with a kernel machine, *Tracing with the Randomly-projected After Kernels* (TRAK) makes progress toward reducing the computational cost of data attribution by reducing dimensionality with random projections and ensembling over independently trained models. However, the method tends to only work well with more than a dozen model checkpoints and a large projection dimension for the model gradients, the storage of which can surpass 80GB when using a ResNet-9 on CIFAR-10. Compared to Datamodels, TRAK gains in runtime are paid for in storage space.

## 4 COUNTERFACTUAL ESTIMATION

We evaluate these approaches and our proposed baseline data attribution for a number of classification examples in computer vision, focusing on datasets such as CIFAR-10 and ImageNet, which are small enough to allow for some comparison with the more expensive approaches of TRAK and Datamodels.

### 4.1 EXPERIMENTAL SETUP

**Training Setup:** We estimate the approximate data removal and data mislabel support for CIFAR-10 and ImageNet. As computing the data support for even a single validation sample requires training multiple models, we restrict ourselves to a reasonably small set of validation samples. We use the same validation samples across all attribution methods. To accelerate the training of these models, we use the FFCV library (Leclerc et al., 2023).

For CIFAR-10 (Krizhevsky et al., 2009), we train ResNet-9 [1] and MobileNetV2 (Sandler et al., 2018) models for 24 epochs using a batch size of 512, momentum of 0.9, label smoothing of 0.1, with a cyclic learning schedule, with a maximum value of 0.5. The test accuracy for these models without any modification to training data is above 92%. We randomly selected 100 validation samples, in

---

[1] https://github.com/wbaek/torchskeleton/blob/master/bin/dawnbench/cifar10.py

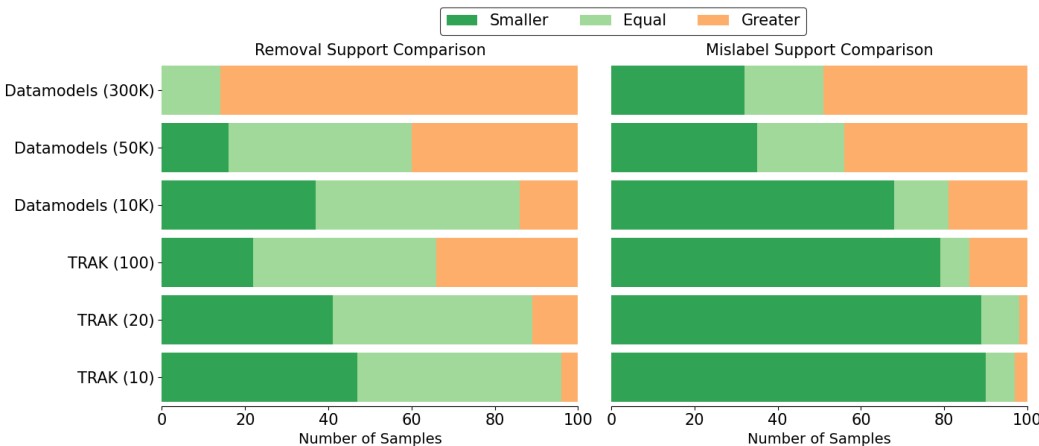

Figure 4: Compared to instances of Datamodels and TRAK, we check whether our data support estimates are smaller, equal, or larger for all 100 validation samples. For 32 samples, our proposed method can find smaller data mislabel support compared to Datamodels (300k models). Even for the data removal case, our approach can find an equivalent support estimate to Datamodels (300k models) for 14 samples.

a class-balanced manner for our brittleness metrics. We remove or mislabel a maximum of 1280 training samples for each validation sample. Our training setup is similar to Ilyas et al. (2022).

For ImageNet (Deng et al., 2009), we train ResNet-18 (He et al., 2015) models for 16 epochs, using a batch size of 1024. We train on $160\times160$ resolution images for the first 11 epochs and increase the training resolution to $192\times192$ for the last 5 epochs. The other hyperparameters are kept the same as CIFAR-10. These models achieve a top-1 validation accuracy of $67\%$. We randomly selected 30 validation samples, from a subset of validation samples that are not misclassified by 4 ResNet-18 models on average. We removed or mislabeled a maximum of 1000 training samples for each validation sample.

**Baselines and Our Setup:** To estimate TRAK scores on CIFAR-10, we train 100 ResNet-9 models and use a projection dimension of 20480. To estimate scores on ImageNet, we train 4 ResNet-18 models and use a projection dimension of 4096. Computing TRAK scores using 4 models already requires 160 GB of storage space, hence we refrain from using a larger ensemble of models.

For Datamodels, we download the pre-trained weights optimized using outputs from 300K ResNet-9 models with $50\%$ random subsets.[2] We also download the binary masks and margins to train our own Datamodels on outputs from 10K and 50K ResNet-9 models, using another 10K models for validation. Since Datamodels are extremely compute-intensive and require training hundreds of thousands of models, we cannot include them as a baseline on ImageNet.

For our baseline approach to train self-supervised models, we use the Lightly library (Susmelj et al., 2020). We train a ResNet-18 model using MoCo (He et al., 2020) for 800 epochs on CIFAR-10, using the Lightly benchmark code.[3] On ImageNet, we download a pre-trained ResNet-50 model trained using MoCo.[4] For our approach, we always use a single model. We denote Datamodels using N models as Datamodels (N), and similarly for TRAK.

## 4.2 CIFAR-10 DATA BRITTLENESS

In Fig. 3, we present the distribution of estimated data removal values for CIFAR-10. Our findings reveal that employing a single model with a MoCo backbone (He et al., 2020) for data removal support proves more effective than employing Datamodels with 10,000 models and TRAK with 20 models. Our approach and Datamodels (10K) identify that 23% samples can be misclassified by

---

[2]https://github.com/MadryLab/datamodels-data

[3]https://docs.lightly.ai/self-supervised-learning/getting_started/benchmarks.html

[4]https://github.com/facebookresearch/moco

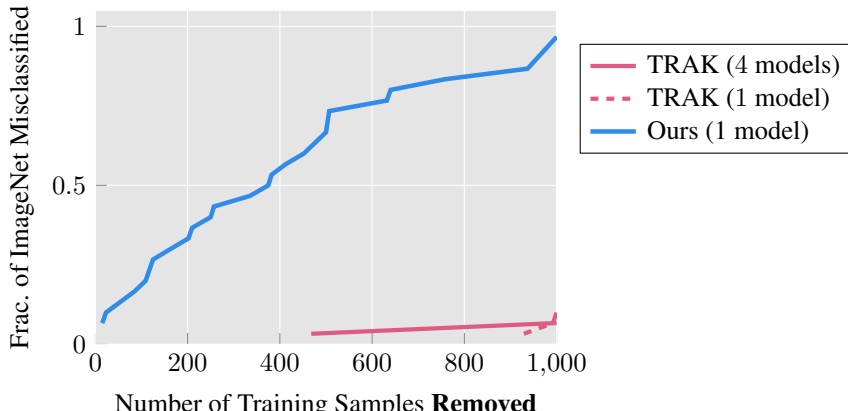

Figure 5: **Our method yields better upper bounds on support size compared to TRAK-4, which requires more storage than the ImageNet dataset itself.** We estimate data removal support for 30 random ImageNet validation samples and plot the CDF of estimates.

removing fewer than 500 (example-specific) training samples while TRAK (20) can only identify 16%. For support sizes up to 1280 images, our approach identifies 55% of validation samples, whereas TRAK (20) and Datamodels (10K) can only identify 28% and 31% samples respectively.

In the same figure, we also depict the distribution of estimated data mislabel support for CIFAR-10. Here, our approach outperforms TRAK (100) and approaches the performance of Datamodels (300K). Here, our approach identifies 47% of CIFAR-10 validation samples that can be misclassified by mislabeling less than 30 training samples! In contrast, TRAK (100) performs poorly identifying only 20% of these samples. DataModels (300K) can identify 50% of validation samples marginally surpassing our performance.

In Fig. 4, we further inspect how well our baseline approach works for each validation sample. We compare the individual estimated support sizes for all 100 samples using our approach versus other baselines. Our results show that for data removal support, across 16% of validation samples, our estimated data removal support is smaller than those of Datamodels (50K). For 44% of the samples our data removal estimates match TRAK and Datamodels (50K). For data mislabel support, our approach finds a smaller support estimate than Datamodels and TRAK for 32% and 79% of the validation samples.

While our baseline approach cannot outperform Datamodels (300K) on data removal, our performance on the data mislabel support is nearly the same. Our baseline approach of using a single self-supervised model can thus serve as a simple, compute, and storage-efficient alternative to estimate data brittleness.

### 4.3 ImageNet Data Brittleness

In Fig. 5, we show our results for data removal on ImageNet. Our results show that for 4 and 16 of the 30 validation samples our estimated data removal support is less than 16 and 130 training samples respectively. In contrast, TRAK (1) and TRAK (4) do not scale well to ImageNet at all and provide much looser data removal estimates. We again emphasize that even scaling to TRAK with 10 models would require around 400 GB of storage space, by our estimate. This highlights the scalability of our baseline approach where a single self-supervised MoCo backbone can provide more accurate data removal estimates than other existing data attribution methods.

### 4.4 Transfer to different architecture

Datamodels and TRAK utilize information tied to the model architecture such as gradients or logits from an ensemble of models. However, different neural network architectures are known to exploit similar biases and output similar predictions (Mania et al., 2019; Toneva et al., 2018). In order to better understand how data may be shaping these biases we test how well attribution scores from

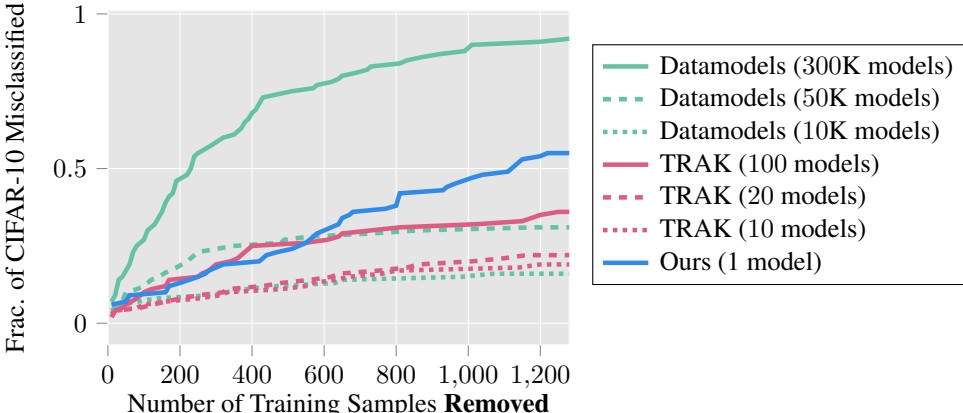

Figure 6: **Our baseline approach is model agnostic and performs well across different architectures.** We evaluate how attribution scores transfer from one architecture transfer to another. We use ResNet-9 scores for TRAK and DataModels and estimate data removal support for MobileNetV2. For our approach, we use the same ResNet-18 backbone.

these approaches transfer to other architectures. Since our approach does not use any information about the model architecture and only leverages the data, we expect our baseline approach to transfer across different architectures.

In Fig. 6, we compare TRAK, Datamodels, and our attribution scores and evaluate them on a MobileNetV2 architecture (Sandler et al., 2018). The results show that our approach using ResNet-18 continues to predict accurate data removal estimates surpassing TRAK (100) and Datamodels (50K), which suffer a large degradation in performance. Datamodels (300K) also suffer degradation in performance but provide tighter estimates than our approach. This suggests that while simply relying on visual similarity may be useful for efficiently predicting counterfactuals, additional biases within the architecture may also have an influence.

## 5  DISCUSSION

### 5.1  ROLE OF VISUAL SIMILARITY

In Fig. 7, we plot the most similar training images according to Datamodels, TRAK, and our method. Given that our approach relies on comparing MoCo features from the same class as the target image, it makes sense that the closest training images are visually similar. On the other hand, the most similar training images found by Datamodels (Ilyas et al., 2022) and TRAK (Park et al., 2023) show more variability. Despite the variability of most similar train images, Datamodels (300K) outperforms all other methods in the counterfactual tasks assessed in Fig. 3, hinting at the importance of additional contributing factors. Still, our method underscores the significant impact of relying solely on visual similarity, essentially showing that a significant fraction of data attribution can be achieved without knowledge of the learning algorithm, based only on knowledge of the training set.

### 5.2  OTHER RELATED WORK

Data attribution methods should produce accurate counterfactual predictions about model outputs. Although a counterfactual can be addressed by retraining the model, employing this straightforward approach becomes impractical when dealing with large models and extensive datasets. To address this problem, data attribution methods perform various approximations.

The seminal work on data attribution of Koh & Liang (2017) proposes attribution via approximate *influence functions*. More specifically, Koh & Liang (2017) identify training samples most responsible for a given prediction by estimating the effect of removing or slightly modifying a single training sample. But being a first-order approximation, influence function estimates can vary wildly with changes to network architecture and training regularization (Basu et al., 2021). Nevertheless,

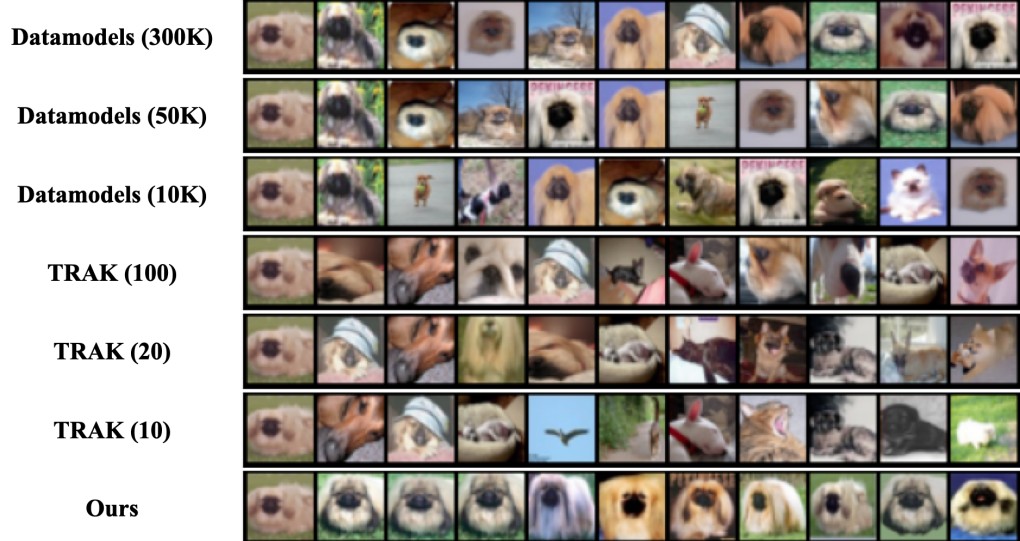

Figure 7: **Our attribution method consistently selects the most visually similar training images by design**. In each row, we plot the same target test image (Index 31), followed by ten most similar training images according to each attribution method.

approximating influence functions is reasonably inexpensive and has recently also been attempted for multi-billion parameter models Grosse et al. (2023).

Measuring empirical influence has also been attempted through construction of subsets of training data that include/exclude the target sample (Feldman & Zhang, 2020). In a related approach, TracIn (Pruthi et al., 2020) and Gradient Aggregated Similarity (GAS) (Hammoudeh & Lowd, 2022a;b) estimate the influence of each sample in training set $S$ on the test example $z_t$ by measuring the change in loss on $z_t$ from gradient updates of mini-batches. While TracIn can predict class margins reasonably well, the method struggles at estimating data support. Other methods for influence approximation include metrics based on representation similarity (Yeh et al., 2018; Charpiat et al., 2019). Another related line of work has utilized Shapley values to ascribe value to data, but since Shapley values often require exponential time to compute, approximations have been proposed (Ghorbani & Zou, 2019; Jia et al., 2019). In general, there seems to be a recurring tradeoff: methods that are computationally efficient tend to be less reliable, whereas sampling-based approaches are more effective but require training thousands (or even tens of thousands) of models.

## 6 CONCLUSION

Data attribution approaches are computationally expensive and can be prone to inaccuracy. While these approaches exhibit promise and capability, their scalability to large-scale models remains uncertain. Our work highlights the importance of visual similarity as a baseline for counterfactual estimation, providing valuable insights into data attribution. Our approach demonstrates scalability and accuracy, particularly in attributions for ImageNet, where it outperforms other state-of-the-art methods while maintaining manageable compute and storage requirements. Remarkably, our approach achieves these results without any reliance on training setup details, target model parameters, or architectural specifics. Our work shows that strong data attribution can be achieved solely based on knowledge of the training set.

## 7 REPRODUCIBILITY

For reproducibility, our supplementary material includes the source code and top-k training indexes from baselines and our method. We will also release the model weights, but these are too large to include in the supplementary. We also discuss our full experimental setup and include links to the baselines we used in Section 4.1. Details regarding our evaluation metrics are described in Section 2 and Appendix A.1. Our experiments utilize randomly selected validation samples from CIFAR-10 (Krizhevsky et al., 2009) and Imagenet (Deng et al., 2009), and we also include indices for these validation samples with our supplementary material.

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

# A APPENDIX

## A.1 COMPUTE TIME AND STORAGE REQUIREMENTS

For our compute time estimates, we use NVIDIA RTX A6000 GPUs and 4 CPU cores. We describe how we estimate the wall-clock time, and storage requirements for each method below -

- **Datamodels:** We only take into account the storage and compute cost of training models. The additional cost of estimating datamodels from the trained models, requires solving linear regression whose computational costs are negligible compared to training the models. For compute and storage requirement estimates, we train 100 ResNet-9 models on random $50\%$ subsets of CIFAR-10 and extrapolate to estimate the training time and storage required for 10,000 and 50,0000 models shown in Fig. 1.

- **TRAK:** We use the authors' original code [5] to train, and compute the projected gradients for CIFAR-10 using ResNet-9 Models using a projection dimension of 20480. For storage requirements, we take into account storage used by model weights, and the projected gradients. The results in Fig. 1, show the compute and storage using 10, 20 and 100 models.

- **Ours:** We use Lightly library [6] benchmark code to train a MoCo model using a ResNet-18 backbone on CIFAR-10 for 800 epochs. The results in Fig. 1 show the wall-clock training time for the model, and extracting the features from CIFAR-10 and the storage requirements for model weights.

To calculate the storage requirements, we factor in the storage space necessary for retaining the trained model weights, as they are essential for computing influence on new validation samples across all attribution methods.

## A.2 ADDITIONAL SELF-SUPERVISED FEATURES

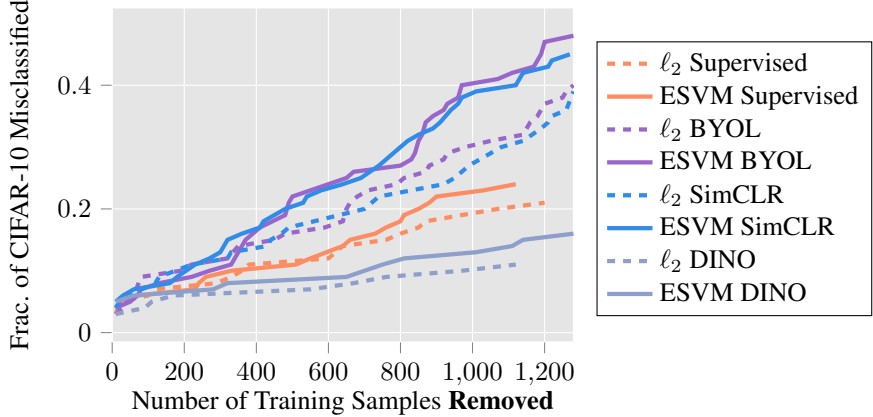

Figure 8: We estimate data removal support for 100 random CIFAR-10 test samples and plot the CDF of estimates.

In addition to utilizing features from MoCo in Section 3, we test our choice of distance function on ResNet-18 features from other self-supervised methods trained on CIFAR-10. In particular, we evaluate BYOL (Grill et al., 2020), SimCLR (Chen et al., 2020), and DINO (Caron et al., 2021) at estimating data removal support in Fig. 8 and mislabel support in Fig. 9. With the exception of DINO, self-supervised features from BYOL and SimCLR outperform the supervised baseline at estimating data removal support. Additionally, we see that in all cases using ESVM distance is more effective than using $\ell_2$ distance to compare features.

---

[5]https://github.com/MadryLab/trak

[6]https://github.com/lightly-ai/lightly

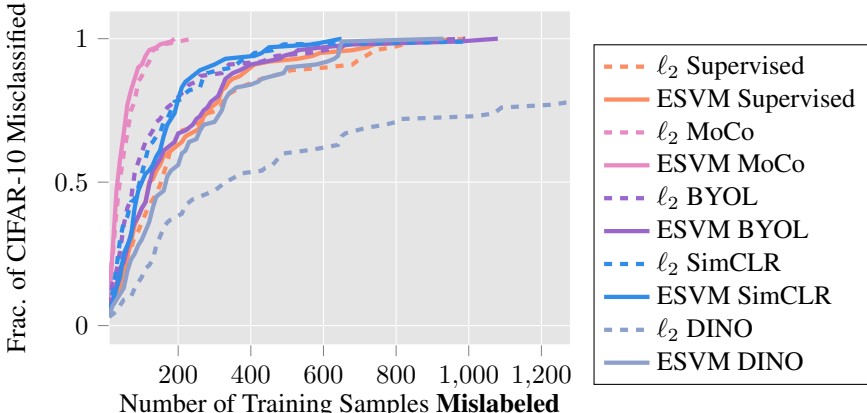

Figure 9: We estimate data mislabel support for 100 random CIFAR-10 test samples and plot the CDF of estimates.

### A.3 ADDITIONAL JUSTIFICATION FOR CHOSEN SUBSET OF TRAIN IMAGES

For a target sample $z_t$, data attribution approaches rank the training samples based on decreasing order of positive influence on $z_t$. For our method, a design choice was whether to rank training samples from all classes or from a selected subset of the training data. One reasonable subset was to select training samples from the same class as the target test sample. In Fig. 10, we show that selecting from the same class is more effective when estimating britteness scores. We maintain this choice for all our experiments.

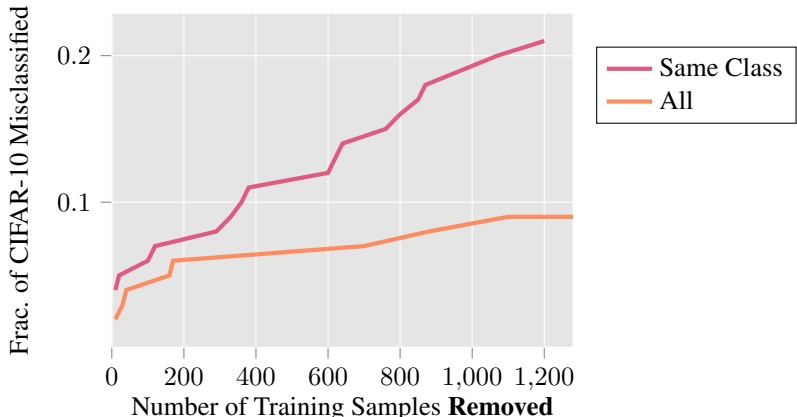

Figure 10: Choosing removal support from all training images is less effective than selecting from the same class as the target image.

### A.4 COMPUTING DATA SUPPORT

We use bisection search to estimate data support. The use of bisection search is supported by the observation that several data attribution approaches are additive (Park et al., 2023), where the importance of a subset of training samples is defined as the sum of each of the samples in the subset. To compute data removal support, we remove $M$ samples (chosen using each attribution method) from the training data and log whether the resulting model misclassifies the target sample. For data mislabeling support, we mislabel $M$ samples (chosen using each attribution method) from the training data and assign a new label corresponding to the highest incorrect logit.

A detailed summary of our bisection search is in Algorithm 1. A key step is CounterfactualTest$(f, S, I_{\text{attr}}[: M])$ which returns the average classification of $N_{\text{test}}$ indepen-

dent training runs where $f_\theta$ is trained on the subset $R = \{z_i | z_i \in S \text{ and } i \notin I_{\text{attr}}[: M]\}$. In other words, for computing data removal support, $f_\theta$ is trained on a subset of $S$ that does not include the first $M$ indices of $I_{\text{attr}}$. For computing mislabeling data support, the only difference is that rather than removing the first $M$ indices of $I_{\text{attr}}$, we relabel those samples with the class of the highest incorrect-class logit, following (Ilyas et al., 2022).

---

**Algorithm 1** Bisection Search for Computing Data Support

---

**Input:** Target sample, $z_t = (x_t, y_t)$
**Input:** Training set, $S$, and a list of top $k$ training set indices $I_{\text{attr}}$ ordered by the attribution method $\tau(z, S)$
**Input:** Model $f_\theta$
**Input:** Search budget, $N_{\text{budget}}$
**Input:** Number of times to test classification, $N_{\text{test}}$
**Output:** $N_{\text{support}}$, size of the smallest training subset $R \subset S$ such that $f_\theta$ misclassifies $x_t$ on average

1: $L \leftarrow 0$
2: $H \leftarrow |I_{\text{attr}}|$
3: $M \leftarrow H$
4: $C_{\text{avg}} \leftarrow \text{CounterfactualTest}(f, S, I_{\text{attr}}[: M])$
5: **if** $C_{\text{avg}} > 0.5$ **then**
6:      **return** -1                                 $\triangleright N_{\text{support}}$ is larger than $k$
7: **end if**
8: $N_{\text{support}} \leftarrow M$
9: **while** $N_{\text{budget}} > 0$ **do**
10:      $N_{\text{budget}} \leftarrow N_{\text{budget}} - 1$
11:      $M \leftarrow (L + H)/2$
12:      $C_{\text{avg}} \leftarrow \text{CounterfactualTest}(f, S, I_{\text{attr}}[: M])$
13:      **if** $C_{\text{avg}} > 0.5$ **then**
14:          $L \leftarrow M$
15:      **else**
16:          $H \leftarrow M$
17:          $N_{\text{support}} \leftarrow \min(M, N_{\text{support}})$
18:      **end if**
19: **end while**
20: **return** $N_{\text{support}}$

---

For bisection search across all attribution methods, we use a search budget of 7. For the CIFAR-10 data brittleness metrics, we aggregate predictions over 5 independently trained models. Thus, to evaluate a single validation sample, we train 35 models (7 budget $\times$ 5 models) for a total of 3500 (35 $\times$ 100 samples) models for a data brittleness metric. On ImageNet, we don't aggregate predictions and only train a single model. Hence, to evaluate a single validation sample on Imagenet, we train 7 models per sample, and a total of 210 models for evaluating a data brittleness metric. Due to the large training cost on ImageNet, we only show results for data removal support. We explicitly point out that these costs are incurred only for analysis of these data attribution methods (see Section 2). Our attribution approach is in comparison, extremely cheap to compute.

## A.5 LINEAR DATAMODELING SCORE

Let $\tau(z, S) : \mathcal{Z} \times \mathcal{Z}^n \to \mathbb{R}^n$ be a data attribution method that, for any sample $z \in \mathcal{Z}$ and a training set $S$ assigns a score to every training sample indicating its importance to the model output. Consider a training set $S = \{z_1, z_2 \ldots z_n\}$, and a model output function $f_\theta(z)$. Let $\{S_1, ..., S_m | S_i \subset S\}$ be $m$ random subsets of the training set $S$, each of size $\alpha \cdot n$ for some $\alpha \in (0, 1)$. The linear datamodeling score (LDS) is defined as:

$$\text{LDS}(\tau(z, S)) = \rho(\{f_{\theta(S_j)}(z) \mid j \in [m]\}, \{\tau(z, S) \cdot \mathbf{1}_{S_j} \mid j \in [m]\}) \tag{1}$$

where $\rho$ denotes Spearman rank correlation (Kokoska & Zwillinger, 2000), $\theta(S_j)$ denotes model parameters after training on subset $S_j$, and $\mathbf{1}_{S_j}$ is the indicator vector of the subset $S_j$. Unlike data brittleness metrics, LDS accounts for samples with positive as well as negative influence.

To compute LDS scores, for our model output function $f_\theta(z)$, we use the correct class margin. This is defined as:

$$f_\theta(z) = (\text{logit for correct class}) - (\text{highest incorrect logit})$$

|  | Models Used | LDS Scores |
|---|---|---|
| Datamodels | 300,000 | 0.56 |
|  | 50,000 | 0.43 |
|  | 10,000 | 0.24 |
| TRAK | 100 | 0.22 |
|  | 20 | 0.15 |
|  | 10 | 0.12 |
|  | 5 | 0.08 |
| Ours | 1 | 0.08 |

Table 1: We compare LDS scores for our approach with other baselines on CIFAR-10. Our proposed approach can perform equivalent to TRAK with 5 models.

Our approach cannot directly be applied to compute LDS scores, as for a validation sample $z_t$ we only focus on training samples with the most positive impact. We propose a simple modification to our approach. We assign a score to each training data based on the inverse of signed $l_2$ distance. The sign is based on whether the label for the training sample matches $z_t$. We then threshold our scores, such that all scores beyond the top-5% are zero leading to sparser attribution scores. The sparsity prior has been shown to be effective for data attribution (Ilyas et al., 2022; Park et al., 2023).

In Table 1, we present a comparison of LDS scores using our baseline approach, TRAK and Datamodels. Although our baseline was not initially designed for direct LDS score approximation, a simple adaptation demonstrates comparable performance to TRAK (5) on CIFAR-10. TRAK with a larger ensemble of models can achieve higher LDS scores. The Datamodels framework was optimized for this objective and trained as a supervised learning task, using tens of thousands of models. Hence, it achieves a better correlation with LDS.

It is important to highlight that while Datamodels and TRAK outperform our baseline in terms of LDS with extensive model ensembles, this metric provides limited insights into understanding machine learning models. Our baseline approach excels in data brittleness metrics, offering a faithful representation of which training samples provide the most positive influence for a test sample.

