# OpenReview forum: "A Simple and Efficient Baseline for Data Attribution on Images"
_ICLR.cc/2024/Conference — ICLR 2024 Conference Withdrawn Submission_

### Official Review · Reviewer_wyU7 · 2023-11-01

**Soundness:** 3 good
**Presentation:** 2 fair
**Contribution:** 2 fair
**Rating:** 5
**Confidence:** 4

**Summary:**

This paper develops a data attribution method, for attributing classification performance of a neural network on a given sample to training samples, that provides a better efficiency-accuracy trade-off compared to existing attribution methods. The proposed method computes the affinity of a given evaluation image with all images in the training set, in the latent space of a pretrained neural network and under a particular metric. The paper provides experiments on CIFAR-10 and ImageNet showing on-par performance with state-of-the-art data attribution methods using only a fraction of their memory and computational budget. The main finding of the paper is that relying only on visual similarity is effective in discovering the smallest set that affects the classification performance of a given image.

**Strengths:**

The paper tackles an interesting problem, namely making data attribution practical, and provides an insightful comparison with state-of-the-art data models. The writing is also clear and easy to follow.

**Weaknesses:**

1- The main scientific finding of this paper is an expected result: that training images with high visual similarity to an evaluation image are important for correctly classifying that image. If this is a surprising finding (the abstract seems to claim it is), the paper should try to emphasize the arguments against it in prior works, and discuss the fault in those arguments. As it stands, I find its main scientific contribution not very significant.

2- While the main practical contribution of the paper, its method, cannot outperform state-of-the-art in the studied datasets, the paper correctly explains that it is more practical. However, the paper does not provide any real-world experiments to show the usefulness of its method in an application. This makes it hard to judge the significance of the proposed method in practice.

3- The paper does not explain its related works in sufficient detail, and as a result, a lot of the methodologies it borrows from prior works is unclear. In particular, a detailed discussion of metrics and their relation to the considered metrics is missing. The Appendix provides some more detail, specially on LDS, which I think must be part of the main paper.

**Questions:**

My three concerns mentioned in the weaknesses section contain my suggestions.

---

> ### Author Response · Authors · 2023-11-18
> **Response to Reviewer wyU7**
>
> We thank the reviewer for the extensive review of our work. We address the questions and comments below -
>
> ### Visual Similarity
> Ignoring how long (and how much storage) it would take to estimate data support using Datamodels and TRAK, we demonstrate in Figure 7 that the training samples with the most influence on the test query are, in most cases, *not* visually similar. Rankings from Datamodel (300K) and TRAK (100) show images of different classes on varying backgrounds and poses. The TRAK paper states that “in computer vision, visual similarity between two images does not fully capture the influence of one on the other in terms of model behavior.” Our work seeks to evaluate that claim.
>
> ### Real-World Application
> Unfortunately, Datamodels and TRAK cannot estimate data support on ImageNet due to the required storage space and compute time, as we comment on in Section 4.3. Our method is practical in the sense that it is tractable on a large-scale dataset like ImageNet. Evaluating brittleness is considered an application in the Datamodels and TRAK paper.
>
> ### Prior Work and LDS Scores
> We apologize the discussion regarding methods from prior works was unclear. We will rewrite the baseline and evaluation section for clarity in the next version of the paper. We will also move LDS scores to the main section of the paper.

---

### Official Review · Reviewer_TdDF · 2023-11-02

**Soundness:** 2 fair
**Presentation:** 3 good
**Contribution:** 2 fair
**Rating:** 3
**Confidence:** 5

**Summary:**

The paper studies (and revisits) the effectiveness of similarity-based baselines for the problem of data attribution---understanding
how training data influence model predictions. The paper introduces two new metrics based on brittleness, and evaluate their baseline relative to recent SOTA methods (datamodels, TRAK) in terms of performance and other cost measures (time, memory).

**Strengths:**

- The paper is very written and placed well within the context of prior work (though there are some misleading claims throughout, which I highlight below).
- The experiments and evaluations are thorough and well documented

**Weaknesses:**

At a high level, I have serious concerns about the claims made in the paper and the way the overall message is presented:

- The paper claims that the model agnosticity of their approach (and similarity based methods) is a feature, but I strongly disagree. Data attribution at a most basic level is about understanding why the given specific model/algorithm behaves, so it *has* to be model dependent. Otherwise, just by definition, the method cannot capture any biases unique to the model/algorithm (and as many prior works show, different models have very different biases, for example CNNs vs ViTs, etc.). The paper acknowledges this point in passing, but I think this needs to be much more prominent.

Now, as the authors acknowledge, it's possible that there is a largely "model-independent" component that can account for model behavior (it's not crazy to think that different DNNs leverage data in similar ways), but I think it's a long jump to conclude this just from the brittleness metrics (I expand on this below).

- The paper also continues to propagate the misunderstanding in some prior works in this area that similarity is same as influence, which is just not true! This is not even true when you consider the simplest case of a linear model: there, the similarity is given by the natural euclidean product, whereas the influence looks and behaves very differently (it involves the inverse of a gram matrix).

- Also, a similarity-based approach cannot readily surface negative influencers, or even assign relative weights to the positive influencers. It's almost misleading to call even call the approach "attribution" when you cannot assign quantitative weights to examples (that reflect their counterfactual importance). Similarity (used directly) can only capture the relative ordering among positive influencers.

- The issue of metric: while I agree that brittleness-based metrics are also informative, it seems misleading to base most of the paper's claims on two new metrics, which do not capture the points I mentioned above (negative influencers and calibration among positive influencers), while delaying discussion of metric (linear datamodeling score) considered in prior works to the Appendix.

In in fact, their evaluation shows that the similarity baseline only achieves an LDS of 0.05 on CIFAR-10, which is hardly significant.
TRAK ([2], App. E.3) using just 5 independently models can achieve an LDS of 0.329. It might be true that at the same level of minimal compute (a single checkpoint), the baseline method outperforms prior methods. But importantly, the proposed approach method cannot improve even with more compute! (prior work[1] indeed shows that more "ensembling" has marginal effect on similarity-based approaches).
So one could claim that at the certain level of budget, the proposed approach is the best performing, but it is very misleading to say throughout the paper that this simple approach also beats SOTA, which is clearly not the case (and only shown in the Appendix).

All of these concerns considered, I think a more reasonable (and less misleading) conclusion of the paper would have been more along:
similarity-based approaches can be an effective baseline; rather than "similarity is all you need" message that seems more prominent and is misleading given the various reasons above.

Other concerns:
- Not sure why space efficiency is a big consideration at all. Storing both embeddings (for computing similarities) and storing projected gradients (for a single checkpoint) both require same order of memory.

[1] Andrew Ilyas, Sung Min Park, Logan Engstrom, Guillaume Leclerc, Aleksander Madry. "Datamodels: Predicting Predictions from Training Data."
[2] Sung Min Park, Kristian Georgiev, Andrew Ilyas, Guillaume Leclerc, Aleksander Madry. "TRAK: Attributing Model Behavior at Scale
"

**Questions:**

Concerns were raised above.

---

> ### Author Response · Authors · 2023-11-18
> **Response to Reviewer TdDF**
>
> We thank the reviewer for the extensive review of our work. We address the questions and comments below -
>
> ### Model Dependent Features
> We agree that algorithm behavior is dependent on a combination of architecture, optimization, and training data. Our goal was to demonstrate how far we could get by analyzing the training data exclusively. We have re-written relevant phrases in the text to emphasize that visual similarity can be effective while emphasizing there are other factors at play. Our work serves to stake out a baseline based only on the training data, and may help serve to compare against other attribution approaches.
>
> ### Similarity and Data Attribution
> Thank you for your extensive review. We believe there is a misunderstanding of what a data attribution method is. Our method is a data attribution method by definition:
>
> “Consider an ordered training set of examples $S = \{z_1, …, z_n \}$ and a model output function $f(z, \theta)$. A **data attribution** method $\tau(z, S)$ is a function $\tau: \mathcal{Z} \times \mathcal{Z} \rightarrow R^n$ that for any example $z \in \mathcal{Z}$ and a training set $S$, assigns a real valued score to each input $z_i \in S$ indicating its importance to the model output $f(z; \theta^{*}(S))$.” [2]
>
> By measuring the distance of a training sample to the decision boundary of the ESVM, our method actually does assign real number values to training samples which help classification of the query sample, as we describe in Appendix A.5. While the method only assigns positive scores to training examples of the same class, it is a function $\tau: \mathcal{Z} \times \mathcal{Z} \rightarrow \mathbb{R}^n$. Calling a zero function a data attribution method would be misleading because it is useless.
> In contrast, our method is competitive for data removal and data mislabel support, which are **counterfactual by definition** [1,2]. Our approach achieves similar scores to computationally expensive methods for data removal support. We achieve among the best AUCs on mislabel support for CIFAR-10 and are the only method to date that can tractably estimate data support on ImageNet.
>
> We apologize this wasn’t clear in our paper, we’ll revise the introduction and relevant phrases accordingly.
>
> [1] Datamodels
> [2] TRAK
>
> ### Negative Influence and LDS
> We’d like to note that the scores mentioned, TRAK (5 models) in Appendix E.3 in fact uses an ensemble of 100 models with 5 independent runs and thus still has a very high storage cost. We expand on the storage cost of TRAK below. A high LDS does not imply the method will be effective at other counterfactual estimation tasks like estimating data support.  At the very least, our results emphasize that attribution metrics of LDS and data support are orthogonal.
> We agree, that our baseline may not show significant gains with more compute and we apologize this wasn't clear in the paper. We'll highlight this limitation in the next version.
>
> ### Space Efficiency
> For each training and test data, TRAK uses a projected gradient of dimension (from up to 4096 to 20480) which is significantly higher storage than MoCo embeddings (128 for CIFAR-10). So, even for a single checkpoint the approach typically requires higher memory. We tested the performance of TRAK-100 with 4096 dimensions and noticed the AUC on data removal support deteriorated, and hence for CIFAR-10 we used 20480 as the default dimension. We'll update the draft to reflect the same.

---

### Official Review · Reviewer_eNuE · 2023-11-03

**Soundness:** 2 fair
**Presentation:** 3 good
**Contribution:** 2 fair
**Rating:** 3
**Confidence:** 4

**Summary:**

This paper focuses on the problem of data attribution, i.e., estimating how individual training data points influence model outputs. The main contribution of this paper is a compute-efficient and storage-efficient data attribution baseline for images.  Specifically, the baseline identifies “important” training data points via visual similarity using self-supervised feature extractors (i.e., distance in embedding space).  The experiments suggest that this baseline can outperform existing data attribution methods in identifying small data-removal and data-mislabel support sizes.

**Strengths:**

- The paper is well-written in that the problem setting is clear and the experiments (figures, plots, etc) are easy to follow. The design choices in the proposed baseline are described in detail as well.
- The experiments show that for standard image classification tasks, visually similarity (measured via distance in some embedding space) can surface training data points with high positive influence on model outputs.

**Weaknesses:**

- “Our work shows that strong data attribution can be achieved solely based on knowledge of the training set.” In general, the effect or influence of a training data point on a model output is a function of the learning algorithm used to train the model. The data attribution scores of a standard ERM classifier trained on CIFAR would be very different from a random CIFAR classifier. The proposed method, however, would output the same attribution scores for the random classifier and an ERM classifier trained on CIFAR. This is an issue for at least two downstream applications.
    - As noted in the paper, data attribution can be used for debugging model biases (https://arxiv.org/abs/2211.12491) where you want to compare data attributions of two learning/training algorithms (e.g., with and without data augmentation) and see how data changes in the learning algorithm change data attributions. This method, however, would output the same data attribution scores for both algorithms, so it cannot be used to compare data attribution scores in general.
    - Another application of data attribution is to identify backdoor attacks in training data (https://arxiv.org/abs/2307.10163), as data attributions of backdoored test examples would rely on backdoored training examples. By relying solely on visual similarity (not a good proxy in this case) and not the learning algorithm, the proposed baseline is unlikely to succeed in identifying visually dissimilar + backdoored training points that have high influence.
- This method only focuses on examples with high positive influence. In general, data attribution methods identify data points with high influence, but also data points with ~zero influence and negative influence. Identifying points with almost no influence can be used to prune the dataset, whereas identifying points with negative influence can be used to understand what in the training dataset causes a model to misclassify a test example. The appendix suggests a heuristic to identify negative influence, but it is unclear if this works as well as the data brittleness experiments because the LDS score is quite low (0.05) and Figure 7 only visualizes the positive influencers. One concrete way to check this would be to identify how many negative influencers need to be removed to make an incorrectly classified test point correct.
- The method makes a strong implicit assumption: The data attribution score of training example j and test example i does not depend on other training examples in the dataset. However, if there are multiple copies of training example j in the dataset, then the influence of each copy is down-weighted. Intuitively, this is because the effect of removing a single copy on the model output is small if there are other copies in the training dataset that aren’t removed. The proposed method does not account for this, so it will not estimate the influence of individual training data points in scenarios like the one above.
- The actual method identifies high-influence datapoints by comparing visual similarity of a test example to other examples in the same class. This implicitly assumes that training data points from other related classes cannot be positive influencers. However, even for standard image classification tasks, it is possible to have training data points with positive influence that do not belong to the same class. Furthermore, given that this heuristic is “critical” (S3.1) it is unclear how one would extend this method to vision tasks (e.g. data attribution for CLIP) that does not have a fixed class set.

**Questions:**

Writing is vague at times; a few examples below. It would be great to get some clarity about these statements:
    - “It is important to highlight that while Datamodels and TRAK outperform our baseline in terms of LDS with extensive model ensembles, this metric provides limited insights into understanding machine learning models.”
    - “Thus, the latter metric [data removal and data mislabel support size] serves as a better proxy [than LDS] for the data attribution method’s usefulness as a debugging tool.”

---

> ### Author Response · Authors · 2023-11-18
> **Response to Reviewer eNuE**
>
> We thank the reviewer for their extensive insights. We provide responses to comments and questions below -
>
> ### Algorithm Independence
> Our proposed attribution approach is indeed algorithm-independent. We think this is interesting, that an algorithm-independent approach can be pushed much further than was claimed. We know this assumption is simplified and doesn’t hold for all applications.
> The intuition behind our approach is that models broadly learn very similar features from the training samples. This has been shown by several works [1, 2]. We agree this limits the downstream applications for example LDS score estimation, as we’ve shown in our work, and other applications as the reviewer stated. However, we’d again highlight that our simple approach works extremely well for data support metrics, a core debugging application claimed to work well in both Datamodels and TRAK.
>
> [1] Li, Yixuan, et al. "Convergent learning: Do different neural networks learn the same representations?." arXiv preprint arXiv:1511.07543 (2015).
> [2] Kornblith, Simon, et al. "Similarity of neural network representations revisited." International conference on machine learning. PMLR, 2019.
>
> ### Focuses on positive influence samples
>
> Indeed, our baseline approach shows better performance on samples with high positive influence. We only tested a simple heuristic to identify -ve and zero influence samples for performance on the LDS score. While we agree our LDS scores are relatively low, we'd again emphasize that our approach is meant to serve as a baseline and shows much better results on data brittleness estimates.  Our results highlight that attribution metrics of LDS and data brittleness may be orthogonal, while the latter may be more easily achievable.  While attribution approaches have been applied for several tasks, at the core these are meant to estimate counterfactuals.  Data removal and data mislabel supports are other forms of counterfactual tasks, as shown in Datamodels and TRAK.
> We'd also like to clarify that our results in the appendix show an LDS score of 0.08, equivalent to TRAK using 5 models higher than the 0.05 number the reviewer stated.
>
> ### Duplication Assumption
>
> We thank the reviewer for pointing out this example. Sample independence is indeed a strong assumption. But for estimating data support, a downstream application which other data attribution methods have been used for, our results indicate that relying on visual similarity can perform analogously, and be tractable for large-scale datasets like ImageNet. We agree in future works, it would be interesting to relax this assumption.
>
> ### Same class as Positive Influences and Extensions to CLIP
> Our assumption that same-class samples are positive influences is indeed simple. Our results however show this assumption works well in practice. Even for Datamodels on CIFAR-10 using 300K models, we find that for 500 of the most +ve samples, more than 250 belong to the ground truth class of the test data. Thus, this assumption while simplified, even holds true for datamodels. We only define our baseline approach for classification, yes. In future work, it would be interesting to extend this to multimodal approaches.
>
> ### Writing is Vague
> We thank the reviewer for pointing this out. We will clarify the relevant phrases in the next version of our paper.

---

### Official Review · Reviewer_n2Gq · 2023-11-07

**Soundness:** 2 fair
**Presentation:** 3 good
**Contribution:** 3 good
**Rating:** 5
**Confidence:** 2

**Summary:**

This paper uses self-supervised models as a feature extractor for data attribution. It matches or outperforms previous baselines (TRAK and data models) on CIFAR and ImageNet while being much more efficient in terms of compute and storage.

**Strengths:**

- Results are comparable to the baselines, while reducing cost of data attribution
- Shows that a straight forward approach that was previously abandoned can be effective by using self supervised models
- Architecture transfer ablation is important
- Presentation is clear

**Weaknesses:**

- Unsure of some of the assumptions made in the paper - see questions
- Lacks quantitative comparison with baselines of samples chosen - could be good to compute some statistics across this approach and the baselines. For example, are the subsets chosen by this approach more similar to or different than TRAK and data models

**Questions:**

1. Why is smallest subset the right thing to do? Isn't it possible for there to be two disjoint subsets of different size that both can cause a misclassification? In this case, shouldn't there be some attribution to samples in both groups?

2. Is there an explanation for why self-supervised extractors work better for attribution, and why DINO is an exception?

3. Subsetting on images of the same class is justified by empirical results, but in ImageNet there are classes that are very similar. How can you be sure that there is no attribution in this case?

---

> ### Author Response · Authors · 2023-11-18
> **Response to Reviewer n2Gq**
>
> We thank the reviewer for analyzing the strengths of our work in using a simple approach to reduce the cost of data attribution. We provide responses to comments and questions below -
>
> ### Smallest subset identification as a proxy task
>
> We focus on the smallest subset since this subset has the highest cumulative positive influence on the test sample, and has been used as a proxy task for measuring the performance of attribution methods on prior work [1, 2]. While several such subsets can exist, the smallest subset provides a simple intuitive explanation and allows us to compare against other attribution methods by directly comparing support sizes. Note that while our baseline approach focuses on finding the smallest subset, it also provides attribution beyond this subset.
>
> ### Self-Supervised Features
> We hypothesized that visual similarity plays a larger role in data attribution than prior works claimed [1, 2]. Visual similarity b/w samples are better captured using self-supervised models compared to supervised models since the former are trained to be invariant towards different augmentations.
>
> While it isn’t clear why DINO performs significantly worse than other SSL methods, we find it also performs worse in terms of kNN test accuracy on CIFAR-10 (results in the table below). We suspect that the DINO recipe we used may not be suitable for small datasets such as CIFAR-10.
>
> | SSL Method | Accuracy |
> ------| -----|
> | Moco | 89.9 |
> | SimCLR | 87.9 |
> | DINO | 84.8 |
>
> [1] DataModels
> [2] Trak
>
> ### Quantitative Comparison
> We highlight that our results already show quantitative comparisons against baselines. In Figure 4 for each validation sample, we compute if our estimated support is smaller than other baselines. We will also add results regarding the intersection of these subsets in the next version of our paper.
>
> ### Subset Class Assumption
> Our assumption is supported by the intuition that to misclassify a sample, we should remove training data from its vicinity that shares the same class. This would help in increasing the density near the test sample, toward a different class. While we cannot be sure if samples from other classes also have a positive impact on the validation sample, this approach is meant to serve as a baseline. As the reviewer stated, we find this simplistic assumption works very well and outperforms TRAK on Imagenet.